# Analysis of the Impact of Electrochemical Properties of Copper-Doped Electrode Membranes on the Output Force of Biomimetic Artificial Muscles

**DOI:** 10.3390/polym15214214

**Published:** 2023-10-25

**Authors:** Yingxin Ji, Keyi Wang, Gang Zhao

**Affiliations:** College of Mechanical and Electrical Engineering, Harbin Engineering University, Harbin 150001, China; zhaogang@hrbeu.edu.cn

**Keywords:** electrode membrane, ultrafine copper powder, BMAMs, electrochemical analysis, output force characteristics analysis

## Abstract

In this study, a biomimetic artificial muscle electroactive actuator was fabricated using environmentally friendly sodium alginate extract. Ultrasonic agitation was employed to embed ultrafine copper powder within a mesh-like structure formed by multi-walled carbon nanotubes (MWCNTs), aimed at reducing the internal resistance of the composite electrode membrane and enhancing its output force performance. Focused gallium ion beam-scanning electron microscopy observations, energy-dispersive X-ray spectroscopy (EDS) analysis, and surface morphology imaging confirmed the successful incorporation of the ultrafine copper powder into the MWCNT network. Additionally, we designed and constructed an output force measurement apparatus to assess the output performance of biomimetic artificial muscles (BMAMs) doped with varying quantities of ultrafine copper powder. Electrochemical testing results demonstrated that the artificial muscles exhibited optimal performance when doped with a mass of 1.5 g, yielding a maximum output force of 6.96 mN, an output force density of 30.64 mN/g, and a peak average rate of 0.059 mN/s. These values represented improvements of 224%, 189%, and 222% compared to the electrode membrane without the addition of ultrafine copper powder, respectively.

## 1. Introduction

In recent years, biomimetic artificial muscles (BMAMs) fabricated from electroactive polymers have garnered substantial attention within the academic community [1]. These synthetic muscles exhibit a distinctive capability to undergo deformation or generate motion when subjected to an external electric field, potentially offering a substantial simplification of conventional mechanical actuation mechanisms [2]. This innovation not only mitigates the intricacies associated with mechanical structures but also holds the promise of enhancing the efficient conversion of electrical energy into mechanical work [3]. Consequently, this technology has garnered significant attention in fields such as bionics, microelectromechanical systems, and flexible intelligent robotics, thereby ushering in novel prospects for advancement within these domains [4,5,6]. The unique flexibility and bending characteristics of BMAMs enable the conversion of conventional rigid actuation into flexible actuation within specific domains, thereby markedly streamlining mechanical architecture [7,8,9]. Moreover, these artificial muscles faithfully replicate the motion characteristics of biological muscles, while their fabrication apparatus remains uncomplicated and cost-effective. One may consider the case of biomimetic fish-like robots, wherein these artificial muscles can serve as propulsive tails, providing the robotic platform with locomotion capabilities [10]. In the realm of flexible intelligent robotics, BMAMs find application in emulating human hand grasping behaviors towards objects [11]. Hence, BMAMs not only exhibit advanced motion capabilities but also herald innovative applications across multiple domains.

Based on existing research, the preparation of electroactive polymers typically relies on synthetically derived materials, including polyacrylic acid (PAA) [12], polystyrene sulfonic acid salts (PSSA) [13], polyethylene terephthalate (PET) [14], and polyvinylidene fluoride (PVDF) [15]. However, these synthetic processes often entail the emission of a substantial volume of harmful substances, exacerbating environmental pollution concerns [16]. Additionally, in the course of utilizing these materials, inhalation by humans may potentially pose health hazards. Consequently, this study has opted for a promising alternative approach by employing a naturally occurring high-molecular-weight compound extracted from seaweed, namely, sodium alginate, for the fabrication of BMAMs [17]. This method not only contributes to environmentally friendly and health-conscious production processes but also effectively harnesses seaweed resources, aligning with sustainability goals. In recent years, multi-walled carbon nanotubes (MWCNTs) have found widespread application in the field of electroactive actuator fabrication due to their unique electrochemical properties and stable chemical characteristics [18]. Introducing MWCNTs during the production of electrodes for the BMAMs can significantly enhance their performance, such as increased output force and faster response times [19]. However, the inherent limitations of MWCNTs result in the issue of elevated internal resistance in the biomimetic artificial muscle electrodes they are used to create. To address this challenge, it is possible to incorporate metallic materials into the electrode membrane during the fabrication process [20]. This way, the mesh-like structure formed by MWCNTs can effectively encapsulate the metallic materials, thereby reducing the internal resistance of the composite electrode membrane.

Copper, a metallic element, finds extensive applications in the fields of science and engineering due to its exceptional electrical conductivity, low electrical resistance, high electron mobility, and remarkable electrochemical stability [21]. These distinctive characteristics render copper an ideal material for the fabrication of biomimetic artificial muscle electrode membranes. In the preparation of such biomimetic artificial muscle electrode membranes, the incorporation of ultrafine copper powder offers significant advantages. This is attributed to copper’s intrinsic excellent electron mobility, which enables the composite electrode membrane composed of copper and MWCNTs to rapidly conduct internal electrons under an applied electric field, thereby significantly reducing the internal resistance within the composite electrode membrane. The high electrical conductivity of copper ensures efficient current transmission, a crucial aspect for electrode membrane performance. Low electrical resistance further minimizes resistive losses, ensuring swift electron traversal through the material and enhancing the efficiency of the composite electrode membrane [22]. Additionally, copper’s high electron mobility contributes to outstanding electron transfer during the electronic transport process, aiding in reducing energy losses and improving electrode response times. In addition to these advantages in electron transport, copper exhibits exceptional electrochemical stability, signifying its ability to maintain outstanding performance under various environmental conditions, which is crucial for the long-term usage of composite electrode membranes. Furthermore, copper, as a commonly available metal, boasts abundant resources and cost-effectiveness, making it a viable choice for widespread applications [23]. These properties collectively position copper with significant prospects for applications in the preparation of biomimetic artificial muscle composite electrode membranes.

Therefore, the performance of composite electrode membranes, particularly the influence of doping ultrafine copper powders of different qualities on their performance, constitutes a significant topic worthy of in-depth investigation. The primary objective of this study is to explore the fabrication of composite electrode membranes for the BMAMs through the incorporation of ultrafine copper powders of varying quantities and to comprehensively examine their output characteristics. Additionally, we conducted electrochemical analyses on the composite electrode membranes to gain insights into their electrochemical properties [24]. By integrating the findings from output characteristics and electrochemical analyses, our aim is to delve into the performance of the composite electrode membranes and its inherent correlations. This research holds the promise of offering valuable guidance and a theoretical foundation for the future design and applications of the BMAMs.

## 2. Materials and Methods

### 2.1. Materials

In the experimental procedure, the chemical reagents employed primarily included sodium alginate (AR, purity 90%, Shanghai Maclin Biochemical Co., Ltd., Shanghai, China), MWCNTs dispersion (JCWCNDM-10, purity > 98%, CNTs diameter 40–60 nm, CNTs content 10%, Chengdu Jiacai Technology Co., Ltd., Chengdu, China), ultrafine spherical copper powder (Cu, purity > 99.9%), glycerol (CP, purity > 99%, C_3_H_8_O_3_, molar mass 92.09, China National Pharmaceutical Group Chemical Reagent Co., Ltd., Shanghai, China), and distilled deionized water prepared in the laboratory. The key experimental apparatus for fabricating BMAMs included a magnetic stirrer (HJ-6B, manufactured by Changzhou Guowang Instrument Manufacturing Co., Ltd., Changzhou, China), a vacuum drying oven (DZF-6090BZ, Shanghai Boxun Industrial Co., Ltd., Shanghai, China), an ultrasonic cell disruptor (LC-JY92-IIN, Shanghai Wuxiang Instrumentation Co., Ltd., Shanghai, China), and a pneumatic hot press (DEX-3020-250H, Shenzhen Dexin Automatic Co., Ltd., Shenzhen, China). The primary measurement instruments employed were an electrochemical workstation (CHI660E, Shanghai Chenhua Instrument Co., Ltd., Shanghai, China), a focused gallium ion beam-scanning electron microscope (FIB-SEM, Tescan Amber, Tescan Orsay Holding, Brno, Czech Republic), and an analytical balance (FA1004, Shanghai Shangping Instrument Co., Ltd., Shanghai, China).

### 2.2. Preparation of BMAMs

BMAMs are fabricated through the thermal pressing assembly of electroactive membranes and electrode membranes, forming a structure akin to a “hamburger”. The preparation process can be divided into three primary steps: 1. Fabrication of electroactive membranes, 2. Preparation of electrode membranes, and 3. Assembly of BMAMs. Figure 1 illustrates an overview of the fabrication process for the BMAMs [25].

In order to expedite the dissolution process of sodium alginate, which exhibits an extremely slow dissolution rate at room temperature, this study employed a water bath heating and stirring method to fabricate the electroactive membrane for the BMAMs. Firstly, a 500 mL beaker was used, into which 200 mL of distilled water was added and placed on a six-neck magnetic stirrer. Subsequently, the temperature of the magnetic stirrer was set to 50 °C to create a water bath heating environment. Simultaneously, a 150 mL beaker was prepared, containing 100 mL of distilled water with a magnetic rotor added, and this was placed inside the larger beaker with the magnetic rotor’s rotation speed set at 800 rpm. Next, 1.5 g of sodium alginate was carefully weighed using an analytical balance and slowly added to the smaller beaker, stirring continuously until complete dissolution, which took approximately 2.5 h. During this process, 3 mL of glycerol was introduced, followed by an additional 0.5 h of stirring. At this stage, the prefabricated solution for the electroactive membrane was obtained. The prefabricated solution was poured slowly into a square glass mold measuring 10 cm × 10 cm. Subsequently, the preheating temperature of the vacuum oven was set to 60 °C, and the vacuum level was adjusted to 0. To ensure the uniform thickness of the electroactive membrane, a leveling ruler was used to level the vacuum oven. Finally, the square mold containing the prefabricated solution was placed inside the leveled vacuum oven for 36 h of drying.

In the process of preparing biomimetic artificial electrode membranes, ultrasonic dispersion technology was employed to ensure the uniform dispersion of ultrafine spherical copper powder within the composite electrode membrane. The specific preparation steps were as follows:

Firstly, 80 mL of distilled water was poured into a 150 mL beaker, and a magnetic stirrer was placed inside, with the rotation speed set at 800 r/min. Subsequently, 0.5 g of sodium alginate was slowly added, and it was completely dissolved using a water bath heating-stirring method, which took approximately 1.5 h. Following this, 20 mL of a 10% mass fraction MWCNT dispersion solution was injected, and thorough stirring was conducted for 30 min. Next, 0.2 g of ultrafine spherical copper powder was added, and stirring continued for another 30 min.

Subsequently, the uniformly stirred premix was transferred to an ultrasonic cell disruptor for ultrasonic dispersion. The working temperature for the ultrasonic dispersion was set at 50 °C, with a working time of 30 min, an ultrasonic time of 10 s, and a time interval of 5 s. After completing the ultrasonic dispersion, the premix was removed, and 3 mL of glycerol was added, followed by stirring for 5 min, resulting in the preparation of a premix for the composite electrode membrane with doped ultrafine spherical copper powder at a mass of 0.2 g.

Finally, the premix was placed in a level-positioned vacuum drying oven. The preheating temperature was set to 80 °C, the vacuum level was set to 0, and the drying time was set to 24 h. Through the aforementioned steps, we successfully prepared an ultrafine copper powder/MWCNT composite electrode membrane with a doping mass of 0.2 g. Using the same preparation process, we also prepared ultrafine copper powder/MWCNT composite electrode membranes with doping masses of 0.5 g, 1 g, 1.5 g, and 2 g.

To fabricate the BMAMs, we employed a heat-pressing technique involving the use of an electroactive membrane and a composite electrode membrane. To prevent the adhesion of the electrode membrane materials to the workbench during each heat-pressing cycle, we placed tin foil on both sides of the heat-pressing machine due to its smooth surface and excellent thermal conductivity. Initially, we positioned the composite electrode membrane on the worktable of a pneumatic heat press and set the working temperature to 40 °C. Subsequently, we adjusted the heat-pressing arm to make slight contact with the composite electrode membrane and set the pressure to 0 MPa, maintaining this condition for 15 min. Next, we placed the electroactive membrane beneath the heat-pressing arm, aligning it with the composite electrode membrane, while applying a downward pressure of 0.1 MPa. By repeating this process iteratively, we successfully assembled the electroactive membrane with the composite electrode membrane, ultimately yielding a biomimetic artificial muscle structure resembling a “hamburger”.

### 2.3. Testing Methods

To comprehensively investigate the impact of composite electrode membranes doped with ultrafine copper powder on the output force of the BMAMs, this study employed a variety of experimental methods and conducted detailed analyses. Initially, a focused ion beam-scanning electron microscope (FIB-SEM) was employed to observe the surface of the composite electrode membranes, followed by energy-dispersive X-ray spectroscopy (EDS) analysis, which confirmed the successful adsorption of ultrafine copper powder onto the surface of the composite electrode membranes. Subsequently, electrochemical analyses were conducted on the composite electrode membranes doped with varying masses of ultrafine copper powder, generating impedance spectra, and determining the specific capacitance of different mass composite electrode membranes through cyclic voltammetry. Following these analyses, a direct current voltage of 4 V was applied to the assembled BMAMs (as shown in Figure 2a), resulting in observed deformations towards the positive electrode (as shown in Figure 2b). Additionally, real-time monitoring of output force was carried out using an analytical balance integrated into a laboratory-designed testing apparatus (as shown in Figure 2c). Each experimental group comprised three samples to obtain average values and minimize the influence of systemic errors. Finally, through a comprehensive analysis of the experimental results, we could deduce the influence of the electrochemical characteristics of the composite electrode membranes on the output force of the BMAMs.

## 3. Results and Discussion

### 3.1. Analysis of FIB-SEM Results

To validate the experimental efficacy of the ultrasonic impact mixing method, we employed FIB-SEM to observe the surface of biomimetic artificial muscle electrodes doped with ultrafine copper powder. Two types of electrodes were chosen for this study: a pure electrode and the other doped with 1.5 g of ultrafine copper powder. Subsequently, we conducted an elemental distribution analysis of C, O, and Cu within the electrode membranes. The results of the energy-dispersive X-ray spectroscopy (EDS) analysis of the electrode membrane surface are presented in Table 1. An in-depth analysis of the spectra and peak intensities of C, O, and Cu elements in the figures allowed us to determine the relative content of these elements at the observation points within the electrode membrane. Furthermore, we observed the distribution of ultrafine copper powder on the sample surface, as illustrated in Figure 3. These data conclusively affirm the capability of the ultrasonic impact mixing method to uniformly embed ultrafine spherical copper powder within the network structure formed by MWCNTs. This outcome provides robust experimental support for our research.

Table 1 presents the mass distribution and atomic percentages of C, O, and Cu elements on the surface of bioinspired artificial muscle electrode membranes for two doping levels. In Figure 3a [25], Cu was not detected, while in Figure 3b, we observed a mass fraction of Cu elements of 55% and an atomic percentage of 19% on the surface of bioinspired artificial muscle electrode membranes doped with 1.5 g of ultrafine copper powder. These observations indicate the successful incorporation of ultrafine spherical copper powder into the internal network formed by MWCNTs within the bioinspired artificial muscle electrode membrane using the ultrasonic impact mixing method. To validate the effectiveness of the ultrasonic impact mixing method, we also employed field emission scanning electron microscopy to observe the surface morphology of bioinspired artificial muscle electrode membranes doped with different masses, aiming to understand the distribution of ultrafine copper powder on their surfaces.

Utilizing FIB-SEM, we conducted an investigation into the surface morphology of biomimetic artificial muscle electrode membranes that had been infused with varying quantities of ultrafine copper powder. In Figure 4, we present micrographs of the electrode membrane surface at a 20,000 times magnification. Notably, due to the inherent metallic sheen of copper surfaces, we encountered specular reflections during the observation process. It is worth mentioning that our experiments utilized ultrafine spherical copper powder, with the white circular shadows observed in the images corresponding to these ultrafine spherical copper particles. Our observations distinctly reveal the substantial impact of different quantities of ultrafine copper powder doping on the microstructure of the electrode membrane surface. When the mass of ultrafine copper powder doping was zero g, no particles were discernible on the electrode membrane surface. However, in the range of 0.2 g to 0.5 g of doping mass, a limited quantity of ultrafine copper powder particles became evident on the electrode membrane surface. As the doping mass of ultrafine copper powder increased to the interval of 0.5 g to 1 g, a marked escalation in the presence of these particles on the electrode membrane surface was observed. Remarkably, when the doping mass reached 1.5 g, the distribution of ultrafine copper powder particles exhibited remarkable uniformity. Nonetheless, with a doping mass of 2 g of ultrafine spherical copper powder, a substantial number of such particles were observed on the electrode membrane surface, albeit without uniform distribution. This phenomenon is attributed to an excessive doping of ultrafine copper powder, surpassing the load-bearing capacity of the internal network structure of MWCNTs.

### 3.2. Analysis of Output Force Testing Results

In order to monitor real-time variations in the output force of the BMAMs, we established a dedicated force measurement system within our laboratory. This system was employed to assess the output force of the BMAMs composed of electrode membranes doped with different masses of ultrafine copper powder. Each biomimetic artificial muscle group underwent testing for a duration of 900 s at intervals of 0.5 s, resulting in a total of 1800 data samples. Subsequently, we input these experimental data into Origin v9.8 software for data fitting, yielding line graphs illustrating the output force of the BMAMs doped with varying masses of ultrafine copper powder (see Figure 5a). To better characterize the response speed of these BMAMs, we calculated the peak force divided by the time required to reach this peak, a ratio referred to as the peak-to-peak average rate. We employed Origin 9.8 software to compute the peak average rate for each curve. Additionally, we determined the ratio of the peak force to the mass of the BMAMs, denoted as the force density. Subsequently, we utilized Origin 9.8 software to generate a column chart within a single figure (as shown in Figure 5b), illustrating the output force densities and peak average strain rates of the BMAMs doped with varying qualities of ultrafine copper powder. This analysis facilitated an examination of their respective trends.

The output force variation curve of the BMAMs doped with different qualities of ultrafine copper powder is presented in Figure 5a. The experimental results indicate that the incorporation of an appropriate amount of ultrafine copper powder significantly enhances the maximum output force of the BMAMs during the fabrication of the electrode membrane. As the mass of ultrafine copper powder increases, the maximum output force of the BMAMs exhibits an initial rise followed by a decline. The electrode membrane without the inclusion of ultrafine copper powder forms BMAMs with a maximum output force of only 2.15 mN. However, when 1.5 g of ultrafine copper powder is doped into the electrode membrane, the maximum output force of the BMAMs reaches 6.96 mN, representing a remarkable increase of 224%. Nonetheless, when the doping mass of ultrafine copper powder is further increased to 2 g, the maximum output force decreases to 6.66 mN. To better characterize the performance of the BMAMs, we introduce two parameters, output force density and peak average rate, the calculated results of which are depicted in the bar chart presented in Figure 5b. Their trends follow a similar pattern to the variation in the maximum output force of the BMAMs, showing an initial increase followed by a decrease. The BMAMs without the incorporation of ultrafine copper powder have output force density and peak average rate values of 10.59 mN/g and 0.018 mN/s, respectively. However, in the case of doping 1.5 g of ultrafine copper powder, these values increase to 30.64 mN/g and 0.059 mN/s, respectively, representing improvements of 189% and 222% compared to the performance of the BMAMs without ultrafine copper powder doping. Nevertheless, when the doping mass of ultrafine copper powder is increased to 2 g, the output force density and peak average rate decrease to 29.10 mN/g and 0.050 mN/s, respectively. This phenomenon is attributed to the excessive mass of doping, which leads to the formation of a mesh-like structure within the electrode membrane, exceeding its load-bearing capacity. Therefore, even with the incorporation of a higher mass of ultrafine copper powder during the electrode membrane fabrication process, it does not improve the mechanical performance of the BMAMs.

### 3.3. Analysis of Electrochemical Testing Results

Based on previous research findings [25,26], enhancing the performance of biomimetic artificial muscle electrode membranes can be achieved through the incorporation of performance-enhancing materials during the electrode membrane fabrication process, with the aim of augmenting electrode membrane functionality. Copper, as a metal material, not only exhibits exceptional electrical conductivity but also boasts outstanding chemical stability. As a result, it finds widespread application in both scientific research and engineering domains. To investigate the influence of ultrafine copper powder of varying qualities on the electromechanical properties of the composite electrode membrane, we conducted electrochemical tests on composite electrode membranes doped with different qualities of ultrafine copper powder using an electrochemical workstation. We obtained electrochemical impedance spectroscopy (EIS) and CV characteristic curves for the composite electrode membranes doped with varying qualities of ultrafine copper powder. Subsequently, through a comprehensive analysis of the relationship between internal resistance, output power density, and peak average rate for the composite electrode membranes, we elucidated the impact of doping with ultrafine copper powder of different qualities on the electromechanical performance of the electrode membrane. These experimental results contribute to a deeper understanding of the influence of doping materials on the performance of biomimetic artificial muscle electrode membranes.

Through the electrochemical testing of composite electrode membranes doped with different masses of ultrafine copper powder, Nyquist plots were obtained for the composite electrode membranes doped with varying qualities of copper. In the Nyquist plots, the vertical axis represents impedance (imaginary part), while the horizontal axis represents resistance (real part). Hence, we employed the coordinates of the intersections between the curves and the X-axis to estimate the internal resistance of the samples for subsequent analysis. The intersections of six curves with the X-axis were selectively magnified, as depicted in Figure 6a. By calculating the intersection values of these curves with the X-axis, we obtained the internal resistance values of the composite electrode membranes doped with different masses of ultrafine copper powder, which were 2.77 Ω, 2.27 Ω, 2.01 Ω, 1.71 Ω, 1.49 Ω, and 1.24 Ω, respectively. From the graph, it is evident that as the doping mass of ultrafine copper powder increases, the internal resistance of the composite electrode membrane significantly decreases. Figure 6b vividly illustrates the trend in the change in internal resistance of the composite electrode membranes doped with different qualities of ultrafine copper powder during the experiment. The experimental results indicate that with an increase in the doping mass of ultrafine copper powder, the internal resistance of the composite electrode membrane gradually decreases. Specifically, within the range of doping mass from 0 to 0.2 g, the internal resistance of the composite electrode membrane decreases rapidly. In the range of 0.2 g to 0.5 g doping mass, the rate of decrease in internal resistance slows down. When the doping mass is in the range of 0.5 g to 2 g, the rate of change in internal resistance remains relatively consistent. Ultimately, when the doping mass reaches 2 g, the internal resistance of the composite electrode membrane reaches its minimum value, representing only 45% of the internal resistance value of undoped ultrafine copper powder and demonstrating a performance improvement of 56%.

In order to investigate the influence of internal resistance in composite electrode membranes on the output force of the BMAMs, we conducted a comprehensive analysis, focusing on the interplay between internal resistance, output force density, and peak average rate. As depicted in Figure 7a, with an increase in the doping mass of ultrafine copper powder, the internal resistance of the composite electrode membrane markedly decreases, while concurrently, the output force density significantly increases, revealing a clear negative correlation between these two parameters. When the doping mass of ultrafine copper powder falls within the range of 0 to 0.5 g, the internal resistance of the composite electrode membrane diminishes, and the output force density rises sharply. However, when the doping mass of ultrafine copper powder ranges from 0.5 g to 1.5 g, the internal resistance of the composite electrode membrane continues to decrease, while the output force density reaches its peak. Upon increasing the doping mass of ultrafine copper powder to a range of 1.5 to 2 g, the internal resistance of the composite electrode membrane continues to decrease, but the output force density begins to decline. Furthermore, the relationship between the internal resistance of the composite electrode membrane and the peak average rate is observed in Figure 7b. As the doping mass of ultrafine copper powder increases, the trend between these two factors still exhibits a clear negative correlation. When the doping mass of ultrafine copper powder ranges from 0 to 1.5 g, the internal resistance of the composite electrode membrane decreases, and the peak average rate reaches its maximum value. However, when the doping mass of ultrafine copper powder reaches 1.5 to 2 g, the internal resistance of the composite electrode membrane, while decreasing, leads to a decline in the peak average rate. This phenomenon can be attributed to the excessive doping of ultrafine copper powder, exceeding the load-bearing capacity of the inner network structure of MWCNTs.

During electrochemical testing, cyclic voltammetry was employed to assess composite electrode membranes prepared with varying doping levels of ultrafine copper powder. The cyclic voltammetry (CV) curves for the composite electrode membranes doped with different masses of ultrafine copper powder are presented in Figure 8a. By utilizing the sample masses, potential windows, and scan rates employed in the experiments, we were able to calculate the specific mass capacitance of each experimental sample, as shown in Figure 8b. Specifically, the specific mass capacitance of the composite electrode membranes doped with different masses of ultrafine copper powder were found to be 15.51 mF/g, 24.55 mF/g, 27.28 mF/g, 27.99 mF/g, 27.89 mF/g, and 27.68 mF/g, respectively. From the graphs, it can be observed that the specific mass capacitance of the composite electrode membranes reaches a peak value when the doping mass of ultrafine copper powder falls within the range of 0.2 to 0.5 g. However, with further increases in doping mass, the specific mass capacitance of the composite electrode membranes shows no significant variation. This suggests that the specific mass capacitance of the composite electrode membranes is not substantially influenced by the doping level of ultrafine copper powder in their composition. This phenomenon can be attributed to the physical properties of metallic copper. As an excellent conductor, metallic copper facilitates rapid electron conduction within its structure, but it does not possess inherent electron storage capabilities. Therefore, even at higher doping levels, ultrafine copper powder exhibits similar capacitive performance, which is unlikely to significantly impact the mechanical properties of the BMAMs composed of the composite electrode membranes.

In summary, in the process of preparing biomimetic artificial muscle electrode membranes, the addition of an appropriate amount of ultrafine copper powder effectively reduces their internal resistance. The underlying reason for this phenomenon lies in the exceptional conductivity of metallic copper, which allows rapid electron conduction within its structure, consequently lowering the internal resistance of the composite electrode membrane and enhancing its conductivity. When subjected to a direct current voltage, the BMAMs composed of the composite electrode membrane exhibit superior performance due to the reduced internal resistance. This superior performance includes higher output power density and faster peak average rates, as a greater number of electrons can be conducted per unit of time. However, it is worth noting that when the doping mass of ultrafine copper powder exceeds 1.5 g, the MWCNT network within the composite electrode membrane reaches its adsorption limit. Therefore, increasing the doping mass beyond 1.5 g does not further enhance the mechanical performance of the BMAMs. Additionally, metallic copper itself does not possess electron storage capability. Consequently, with an increase in doping mass, there is no significant change in the specific mass capacitance of the composite electrode membrane.

## 4. Conclusions

In conclusion, this study employed an ultrasonic impact mixing method in the preparation of biomimetic artificial muscle electrode membranes, incorporating varying quantities of ultrafine spherical copper powder to fabricate ultrafine copper powder/MWCNT composite electrode membranes. An energy-dispersive spectroscopy analysis revealed that when the doping mass of ultrafine copper powder reached 1.5 g, the mass fraction of Cu elements reached 56%, with an atomic percentage of 19%. Additionally, we conducted field emission scanning electron microscopy observations to examine the surface morphology of the composite electrode membranes. These experimental results convincingly demonstrated the effective embedding of ultrafine spherical copper powder into the internal network formed by MWCNTs using the ultrasonic impact mixing method. Given the limited load-bearing capacity of the internal network structure of MWCNTs, the biomimetic artificial muscle composed of the composite electrode membrane exhibited optimal output performance when the doping mass of ultrafine copper powder was 1.5 g. The maximum output force reached 6.96 mN, resulting in an output force density of 30.64 mN/g and a peak average rate of 0.059 mN/s. Compared to the electrode membrane without the addition of ultrafine copper powder, the performance improved by 224%, 189%, and 222%, respectively. The addition of ultrafine copper powder during the electrode membrane fabrication process significantly reduced internal resistance. When the doping mass of ultrafine copper powder was 1.5 g, the internal resistance decreased to 1.49 Ω, representing a 56% improvement relative to the electrode membrane without ultrafine copper powder doping. Furthermore, a comprehensive analysis revealed a negative correlation between the internal resistance of the composite electrode membrane and its output performance. Finally, an analysis of the cyclic voltammetry (CV) characteristics of the composite electrode membrane indicated that, due to the inherent inability of metallic copper for electronic storage, there was no significant change in specific mass capacitance as the doping mass increased.

## Figures and Tables

**Figure 1 polymers-15-04214-f001:**
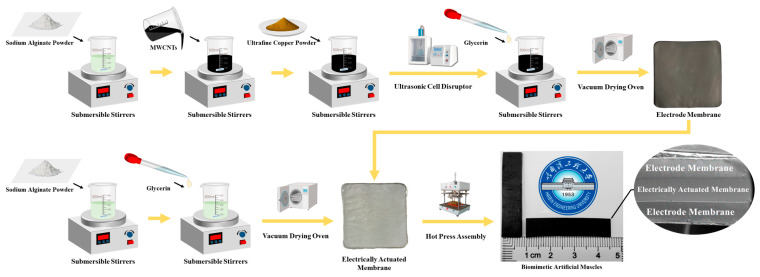
Preparation process diagram of the BMAMs.

**Figure 2 polymers-15-04214-f002:**
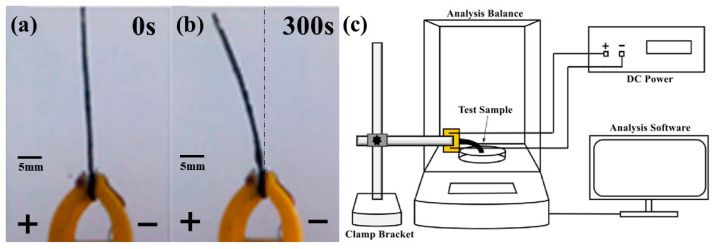
The BMAMs undergo bending to generate an output force. (**a**) The initial state of the BMAMs upon the application of voltage; (**b**) The phenomenon of bending in the BMAMs upon the application of DC voltage; (**c**) schematic diagram of the in-house constructed output force testing apparatus in the laboratory.

**Figure 3 polymers-15-04214-f003:**
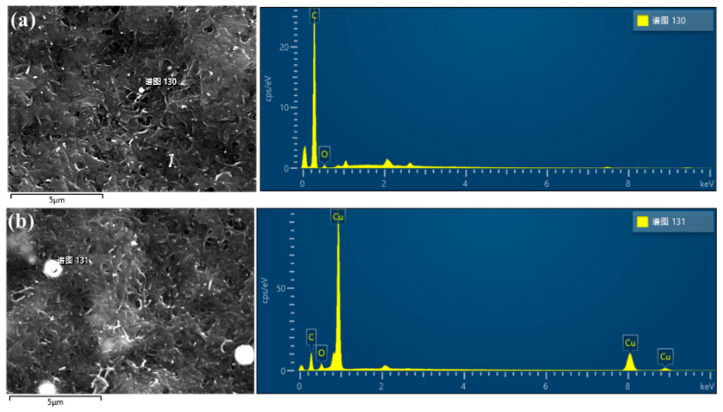
EDS results of electrode membranes with ultrafine copper powder. (**a**) EDS testing results for undoped ultrafine copper powder; (**b**) EDS testing results of Cu with 1.5 g doping mass. (The term “谱图” in the figure refers to “Spectrum”).

**Figure 4 polymers-15-04214-f004:**
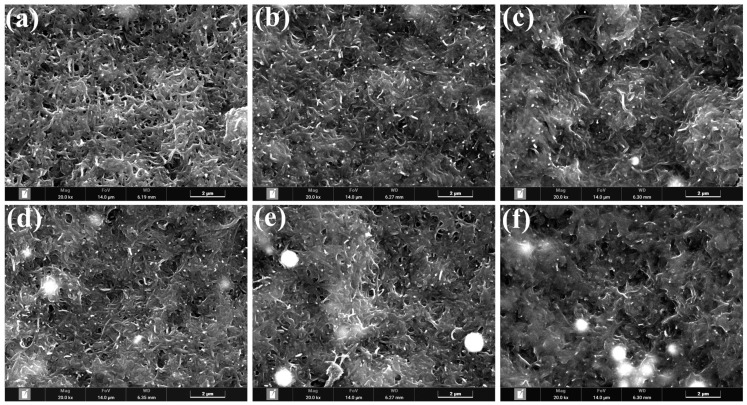
FIB-SEM images of electrode membranes doped with different mass fractions of ultrafine copper powder, with a magnification of 20,000 times: (**a**) 0 g; (**b**) 0.2 g; (**c**) 0.5 g; (**d**) 1 g; (**e**) 1.5 g; and (**f**) 2 g.

**Figure 5 polymers-15-04214-f005:**
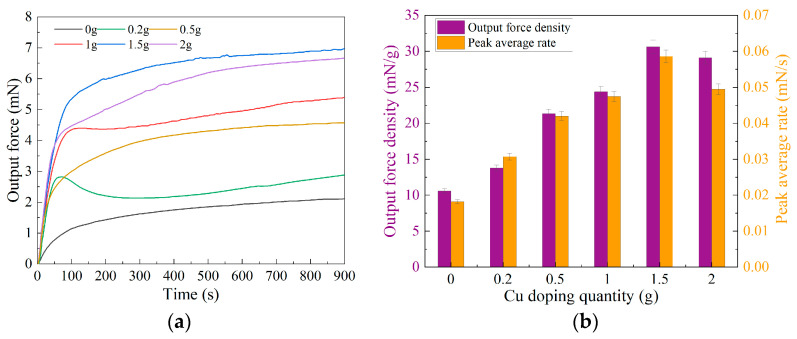
Analysis of experimental results for the output force of the BMAMs: (**a**) graph depicting the output force curve of the BMAMs; (**b**) output force density and peak average rate of the BMAMs comprising electrode membranes doped with ultrafine copper powder of varied masses.

**Figure 6 polymers-15-04214-f006:**
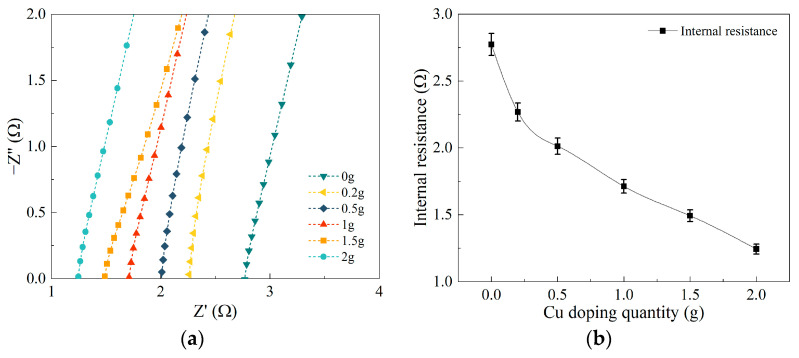
EIS testing results of the BMAMs: (**a**) Nyquist local amplification diagram; (**b**) variations in internal resistance of composite electrode membranes containing ultrafine copper powder.

**Figure 7 polymers-15-04214-f007:**
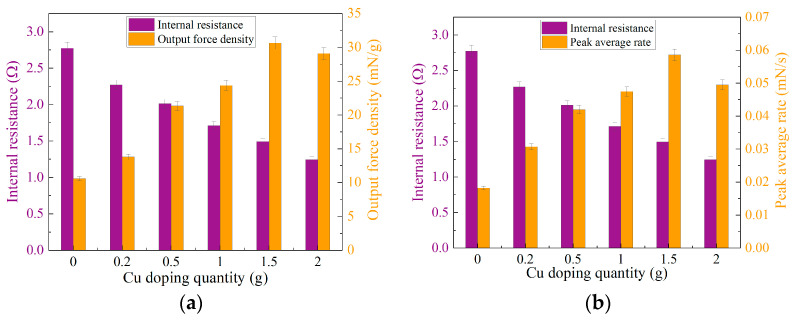
Relationship between the internal resistance of composite electrodes and output performance in academic discourse: (**a**) relationship between internal resistance and output force density; (**b**) relationship between internal resistance and peak average rate.

**Figure 8 polymers-15-04214-f008:**
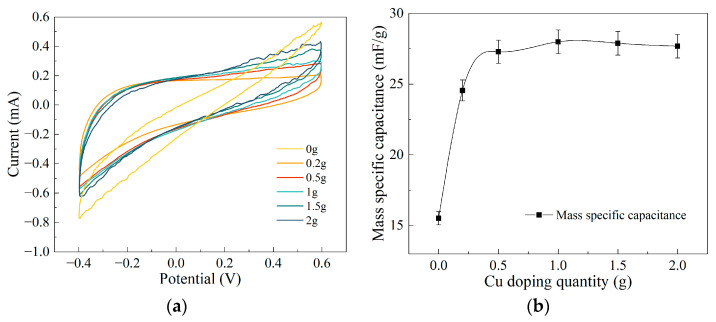
Results of electrode membrane cyclic voltammetry testing: (**a**) CV curve of the electrode membranes; (**b**) the mass specific capacitance of the electrode membranes.

**Table 1 polymers-15-04214-t001:** Distribution of surface elements on two types of electrode membranes.

Element	Position 1		Position 2	
	Mass (%)	Atomicity (%)	Mass (%)	Atomicity (%)
C	96.7	97.02	40.99	75.75
O	3.93	2.98	3.51	4.86
Cu	0	0	55.50	19.39

## Data Availability

The data used for the research described in this manuscript are available upon request.

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
