# Peer review of "Analysis of the Impact of Electrochemical Properties of Copper-Doped Electrode Membranes on the Output Force of Biomimetic Artificial Muscles"

_polymers, 2023, doi:10.3390/polym15214214_

Round 1

Reviewer 1 Report

The article is devoted to the study of Electrochemical and Output Force Characteristics of MWCNTs based membrane doped with Cu nanoparticles.

There are a number of quite serious shortcomings in the article. Among them I can name a few:

1. Insufficient justification for the choice of copper nanoparticles. Why are gold ones worse, for example?

2. Incorrect expression "ultrafine spherical copper powder" (line 212). Powder cannot be spherical; nanoparticles are spherical.

3. Incorrect: "one without any doping of ultrafine copper powder (mass = 0g)" ((line 214). It is more reliable to write - pure (or initial) electrode (MWCNT)..

4. Too big and overly detailed "Materials and methods" part, that makes the article difficult to read. I would suggest moving the finer details to a supplementary part.

5. The article mentions a dual-beam microscope with a gallium source, but only uses an electron gun.

6. Figure 1 containes the same microphoto of membrane as in previos articles of authors (devoted to MoS2 particles). That is strange and incorrect.

7. It is not at all clear how spectral microanalysis is performed. As far as one can judge, the analysis was done at the point. In the second experiment, the point was chosen precisely on a copper nanoparticle. Therefore, the result seems fundamentally incorrect and does not reflect the distribution of nanoparticles, but only allows us to confirm that the nanoparticle is copper. 

8. It is not clear how the mass distribution was calculated, given that the EDS measurements were made incorrectly.

9. What means dots 130 and 131 on Fig. 3, made by Chinese characters?

10. line 457 ("Please add"... funding). What does it means?

11. The value of measurement accuracy, data accurate to hundredths of a percent, raises questions (for example, in the abstract).

To my regret, I do not see new high-quality scientific results regarding the article previously published by the authors on MoS2.

The quality of English is quite good.

Reviewer 2 Report

MS „Y. Ji, K. Wang and G. Zhao: “Analysis of the Electrochemical and Output Force Characteristics of Biomimetic Artificial Muscles …” (polymers-2646991-peer-review-v1) reports results as a continuation (may be parallel work, polymers-2542752) of characterization of biomimetic artificial muscle (BMAM) preparation based on sodium alginate and mesh-like structure of MWCNT.

 I still confirm that the topic is very important in academic and industrial laboratories, and, consequently, can be interesting for potential readers. I was happy to read this second MS (as well). I think, the MS is written well, the experimental design is correct, and the authors used sophisticated technologies in addition to routine laboratory procedures. The structure and the readability of the MS are good, I have only few minor comments for taking into consideration before publishing it.

 1)      Is it possible to shorten the title? I usually prefer shorter titles.

2)      L.96.: „...ultrafine copper powders of varying qualities...” – „qualities”? Not „quantities”?

3)      L.110.: I would use the appropriate indices in chemical structures in a scientific paper.

4)      It is strange that there is no literature reference in the whole Materials and methods section, however, the work is based on several former reports.

5)      In Table 1.: What is the meaning of „Position 1” and „Position 2”?

6)      The expression „ultrafine spherical copper” appears twelve times in rows 246-261.

7)      Do we need (can we do) 4 and 5 digits accuracy for output force density and peak average rate? Especially, if errors were not indicated (by the way). Same for internal resistance.

I am not native English speaker, I am not qualified to evaluate the English quality of the MS. However, I found the readability of the MS good, no issue is detected.

 Final conclusion: I suggest the MS accepted for publication with minor revision.

Round 2

Reviewer 1 Report

I suppose after made corrections the manuscript can be published.

As far as i see the English language is good enough.

Reviewer 2 Report

 MS “Y. Ji, K. Wang and G. Zhao: Analysis of the Impact of Electrochemical Properties of Copper Doped Electrode Membranes on the Output Force of Biomimetic Artificial Muscles (polymers-2646991-peer-review-v2) is the revised version of the MS submitted with slightly modified title (polymers-2646991-peer-review-v1) by the same authors.

 MS is now suitable for publication, I have only two comments for consideration.

 1)    I would avoid Chinese character in the text.

2     2)    I understand the meaning of “Position 1” and “Position 2”. Could you, please, make it clear also for the readers?